# Development of Improved High-Performance Liquid Chromatography Method for the Determination of Residual Caprylic Acid in Formulations of Human Immunoglobulins

**DOI:** 10.3390/molecules27051665

**Published:** 2022-03-03

**Authors:** Adela Štimac, Tihana Kurtović, Nediljko Pavlović, Beata Halassy

**Affiliations:** 1Centre for Research and Knowledge Transfer in Biotechnology, University of Zagreb, Rockefellerova 10, 10000 Zagreb, Croatia; tkurtovi@unizg.hr; 2Institute of Immunology, Inc., Rockefellerova 10, 10000 Zagreb, Croatia; npavlovic@imz.hr

**Keywords:** caprylic acid, high-performance liquid chromatography method, convalescent anti-SARS-CoV-2 human plasma, human immunoglobulins, validation, quality control

## Abstract

Quality control of human immunoglobulin formulations produced by caprylic acid precipitation necessitates a simple, rapid, and accurate method for determination of residual caprylic acid. A high-performance liquid chromatography method for that purpose was developed and validated. The method involves depletion of immunoglobulins, the major interfering components that produce high background noise, by precipitation with acetonitrile (1:1, *v*/*v*). Chromatographic analysis of caprylic acid, preserved in supernatant with no loss, was performed using a reverse-phase C18 column (2.1 × 150 mm, 3 μm) as a stationary phase and water with 0.05% TFA–acetonitrile (50:50, *v*/*v*) as a mobile phase at a flow rate of 0.2 mL/min and run time of 10 min. The developed method was successfully validated according to the ICH guidelines. The validation parameters confirmed that method was linear, accurate, precise, specific, and able to provide excellent separation of peaks corresponding to caprylic acid and the fraction of remaining immunoglobulins. Furthermore, a 2^4−1^ fractional factorial design was applied in order to test the robustness of developed method. As such, the method is highly suitable for the quantification of residual caprylic acid in formulations of human immunoglobulins for therapeutic use, as demonstrated on samples produced by fractionation of convalescent anti-SARS-CoV-2 human plasma at a laboratory scale. The obtained results confirmed that the method is convenient for routine quality control.

## 1. Introduction

Passive immunotherapy involves the administration of antibodies collected from individuals who have recovered from an infection, or have been vaccinated against it, to a patient susceptible to the disease in question. Today, when the whole world is faced with the COVID-19 pandemic induced by the virus named SARS-CoV-2, passive immunotherapy has once again been drawn to attention as the promising treatment option, together with development of refinement strategies for feasible production of effective and safe immunoglobulin G (IgG) preparations [1,2,3,4].

Pure IgGs are extracted from pooled plasma collected from convalescent donors by sequence of different fractionation steps, among which the most widely used are precipitation and liquid chromatography [5,6,7,8,9,10]. Caprylic (octanoic) acid (CA) as a fractionation agent is frequently used to purify IgGs from both animal [11,12,13] and human plasma [4,14,15,16,17], particularly in recent developments of SARS-CoV-2 specific immunoglobulin preparations [4,18,19]. CA selectively precipitates non-immunoglobulin proteins without affecting IgGs, which remain in solution, preserving their structural and conformational stability [6,7,20,21,22]. The mechanism of CA-based precipitation is not yet fully understood. Morais and Massaldi proposed that CA binds to the specific sites on the proteins, thus increasing the hydrophobicity of their interfacial surfaces and, consequently, protein–protein interactions, which ultimately leads to the precipitation [23].

Residual CA has been considered a process-related impurity; thus, it has to be removed from the final product, and its absence has to be reliably documented. Determination of CA traces has usually been performed by high-performance liquid chromatography (HPLC), gas chromatography, and colorimetric methods [11,24,25,26,27]. Herrera et al. [11] described a reverse-phase (RP) HPLC method that was applicable for the antivenom production process. However, the precise and accurate measurement of CA in a wider concentration range was needed, as we noted during the development of a refinement strategy for fractionation of convalescent anti-SARS-CoV-2 human plasma. Moreover, it was necessary to modify the sample preparation due to the observed loss of CA during the diafiltration process.

Therefore, in this study, we present an improved RP HPLC method for the determination of residual CA in formulations of human IgGs. For its validation, quality control (QC) samples were prepared by spiking of the intravenous immunoglobulin (IVIG) product with a known amount of CA. A procedure for depletion of the interfering IgG fraction from QC samples and elimination of background noise was optimized. The application of developed and validated method was tested on samples obtained before and after the diafiltration step from the CA-mediated protocol for downstream processing of convalescent anti-SARS-CoV-2 human plasma that was established on a laboratory scale.

## 2. Results and Discussion

### 2.1. Development of HPLC Method and Sample Preparation

The aim of this work was to develop a sensitive RP HPLC method that would be suitable for determination of residual CA in samples of IgGs purified from convalescent human plasma, complying the principles of good laboratory practices. HPLC analysis of CA was carried out using an RP C18 column. Different chromatographic conditions were examined, while 0.05% TFA in water–acetonitrile (ACN) = 50:50 (*v*/*v*) as the mobile phase and a flow rate of 0.2 mL/min were chosen as optimal concerning resolution and separation of CA and IgGs (Table 1). Although IgGs present in samples do not interfere with CA, they produce a background noise in analysis. Even at concentrations as low as 1 mg/mL, the intensity of peaks corresponding to IgGs is much stronger than that produced by residual CA (Appendix A). Considering that expected concentrations of IgGs in fractions from human plasma processing would be at least fivefold higher, elimination of the background noise in QC samples was a necessary prerequisite prior validation and, later on, implementation of the developed method for the quality control. According to the literature, interfering IgGs could be removed by ultrafiltration in a centrifugal filter unit of 10 kDa nominal molecular weight limit [11] or by their precipitation with ACN [14] or ACN/trifluoracetic acid (TFA) [24].

Therefore, for efficient elimination of IgGs from QC samples, diafiltration and precipitation of IVIG with various solvents were examined. First, diafiltration in a centrifugal filter unit of 100 kDa nominal molecular weight limit (Vivaspin^®^ 20 mL 100 kDa PES) was attempted, but recovery of CA in filtrate was only about 75%. In order to determine possible causes of partial loss of CA during diafiltration process, its amount in retentate and filtrate of QC samples (10 mL), as well as in samples obtained by rinsing the centrifugal filter unit three times with water (10 mL), was measured. The results revealed that the filtrates had about 75% CA, and the retentates had 2–4% CA (Figure 1). The remaining CA very likely bound to the membrane of centrifugal filter unit, given that it could be released and washed by rinsing with water. The concentration of CA in filtrates obtained by rinsing (filtrates 1, 2 and 3) decreased after each cycle of diafiltration. Moreover, the rinsing efficiency was higher in sample with the lowest CA concentration (300 ppm), which can be explained by the higher solubility of CA in water (the solubility of CA in water is 0.068 g in 100 mL, 750 ppm). In order to achieve complete recovery, it would be necessary to perform additional rinsing cycles, but then the CA concentration in the filtrates would be below the LOQ value of the method.

Therefore, precipitation of IgGs with ACN (with or without TFA), methanol, and ethanol was investigated. Addition of one volume of acetonitrile almost completely precipitated IgGs, in contrast to methanol and ethanol, which appeared completely ineffective. The addition of TFA in ACN did not bring any improvement to precipitation; thus, ACN was selected as the optimal precipitating agent. Furthermore, although an increase in ACN content from one to two volumes had a beneficial impact on IgG precipitation in QC samples, it resulted in over-dilution of CA traces, thus decreasing the sensitivity of the assay. Therefore, before HPLC analysis, IgGs in QC samples were precipitated by ACN (1:1, *v*/*v*) and removed by centrifugation. The obtained supernatants gave a convenient separation profile, as well as a high recovery and reproducibility of CA in QC samples.

### 2.2. Validation

The developed HPLC method was validated in terms of specificity, linearity, precision, accuracy, limit of detection (LOD), limit of quantification (LOQ), and robustness, according to International Conference on Harmonization (ICH) guidelines [28].

#### 2.2.1. Specificity

The method specificity refers to the ability to accurately measure the analyte response in the presence of all potential matrix components. The amount of IgGs, retained in QC samples after precipitation with ACN, was sufficiently low and did not interfere with CA quantification by the HPLC method, as demonstrated by complete separation of the CA (RT = 8.4 min) from the IgG (RT = 2.1 min) peak (Figure 2). Therefore, the method was confirmed as specific and suitable for CA quantification in QC samples.

#### 2.2.2. Linearity

The linearity of the method was evaluated by analyzing seven different concentrations of CA standard solutions, each run in duplicate. The calibration curves were obtained by plotting the peak area versus the amount of CA by the least-square linear regression analysis. The method was found to be linear in the range of 25–2500 ppm CA (22.75–2275 μg/mL) and appropriate for measuring the amount of CA in samples from fractionation of convalescent anti-SARS-CoV-2 human plasma (Appendix A). This range of CA concentrations is much wider than the range reported by Herrera et al. (400–600 μg/mL) [11]. The *R*^2^ values for the calibration curves obtained from analyses independently performed over three consecutive days were found to be greater than 0.999, indicating a linear relationship between the amount of CA and peak area. ANOVA analysis (Table 2) proved that there was linear regression with no deviation from linearity (*p* < 0.05).

#### 2.2.3. Precision and Accuracy

Precision is the closeness of a series of measurements obtained from multiple sampling of the same sample, and it was determined by measuring repeatability (intra-day) and intermediate precision (inter-day) over three consecutive days. The intra-day and inter-day precision of the proposed method was measured by analyzing three concentration levels of QC samples, each in hexaplicate, on three separate days. Method precision was expressed in terms of percentage RSD. The values of the percentage RSD obtained for intra-day and inter-day precision measurement were in the range of 0.457–0.848% and 0.746–1.069%, respectively (Table 3). All obtained percentage RSD values (<2%) confirmed the precision of the method.

The accuracy of the method is the closeness of the measurement between the true and the found value, and it is usually determined by spiking known amounts of standard into the sample. The accuracy of the developed method was determined by calculating the percentage recovery of CA in QC samples spiked with CA at three concentration levels, each in hexaplicate. The amount of CA was calculated by applying obtained values to regression equations. The percentage recovery of CA (intra- and inter-day) was found to be in the range of 98.58–101.02% (Table 3), which confirms the accuracy of developed method.

Herrera et al. [11] reported that recovery of CA was between 98.8% and 100.7%, and that intra-day precision was <2.1%. Concerning recovery, we obtained comparable values. In our study, even lower intra-day precision was achieved (<0.85%). Since they did not evaluate inter-day precision, a comparison is not feasible.

#### 2.2.4. Limit of Detection (LOD) and Limit of Quantification (LOQ)

The LOD of an analytical method is the lowest analytical concentration at which an analyte can be detected qualitatively, while LOQ is the lowest concentration of the analyte that can be quantitated with acceptable precision. According to ICH guidelines [28], the LOD and LOQ may be determined by visual evaluation, signal-to-noise ratio (S/N), or calculation based on the standard deviation (SD) of the response and slope of the calibration curve. It was shown previously that LOD and LOQ can differ depending on the used method (visual evaluation, S/N ratio, or the calibration line method). Even within the calibration line method, quite different results can be obtained, depending on whether the SD of the regression line or the SD of the *y*-intercept is used [29,30,31,32].

In the present study, the LOD and LOQ values of the developed method were determined on the basis of both the S/N ratio and the calibration curve using samples containing CA at low concentrations (Table 4). The LOD and LOQ values were calculated with two different calibration curve methods according to LOD = 3.3 *σ*/*S* and LOQ = 10 *σ*/*S*. In the first method, *σ* is the residual SD of a regression line; in the second method, *σ* is the SD of the *y*-intercept.

LOD and LOQ values determined by all three methods were similar and proved the sensitivity of the developed HPLC method. Furthermore, obtained values were lower than those obtained by Herrera et al. [11]. They determined a value of 13.46 μg/mL for LOD and a value of 44.85 μg/mL for LOQ, while our obtained values were 8 ppm (7.28 μg/mL) for LOD and 25 ppm (22.75 μg/mL) for LOQ. Alonso et al. [17] and Valdenberg et al. [18] reported that their IgG products contained less than 8 μg/mL CA, albeit without specifying the method; hence, a comparison is not possible. The method presented here would obviously be suitable for the control of CA impurities, being equally sensitive.

#### 2.2.5. Robustness

The robustness examines the reliability of an analysis with respect to slight variations in method conditions and parameters [28,33]. Selected variations in this method included pH ±0.2 units, wavelength ±2 nm, ACN solvent ratio (*v*/*v*) in the mobile phase (%ACN) ±5%, and flow rate ±0.02 mL/min. The effect of selected operating factors (pH, wavelength, %ACN, and flow rate), each at two levels, was assessed using four-factor fractional factorial design (2^4−1^), requiring the performance of eight experimental runs. The evaluation of responses measured in a robustness test for chromatographic methods includes quantitative aspects responses (e.g., area) and system suitability test (SST) responses (e.g., resolution) [28,33]. The retention time (RT), peak area (A), theoretical plate number (N), resolution (R), capacity factor (k’), and tailing factor were chosen as response variables and were determined in all experimental runs (Table 5). According to Food and Drug Administration recommendations [34], a system is suitable when values for N, R, k’, and tailing factor are >2000, >2, >2, and ≤2, respectively. All obtained values of SST responses were within the recommended values, which indicated good column efficiency, good resolution, good retention on the column, and symmetry of the peak.

The evaluated simple effects (E) of four selected operating factors on each of six observed responses indicated that percentage ACN induced the highest, mostly negative shift in monitored responses (Table 6).

In order to identify statistically significant effects, standardized effects were estimated as the ratio of simple effect and standard error of an effect. The standardized effects of examined operating factors on monitored responses are presented in a Pareto chart (Figure 3), which demonstrates the absolute values of the standardized effects and plots a reference line to indicate which effects are statistically significant (*p* = 0.05). It was observed that both the percentage can content in the mobile phase and the flow rate had significant negative effects on retention time, peak area, theoretical plate number, and resolution, which means that an increase in the effect factor level decreased the response value. Moreover, the percentage ACN content by itself had a significant negative effect on the capacity factor and positive effect on the tailing factor. Hence, it was identified as the most crucial factor, impairing all responses, followed by flow rate. The pH had only a small positive effect on peak resolution. The wavelength had no effect on any response. Therefore, the percentage ACN content in the mobile phase and flow rate should be strictly regulated during method performance.

### 2.3. Application of Developed Method

The suitability of the developed method for the routine quality control was examined by employing IgG fractions from downstream processing of convalescent anti-SARS-CoV human plasma. The refinement strategy was developed at a laboratory scale to provide a safe and effective treatment option for COVID-19. Briefly, IgGs were obtained by precipitation with 5% CA (crude IgG) and were subsequently diafiltrated for CA removal (pure IgG) using a centrifugal filter unit of 100 kDa nominal molecular weight limit. In order to achieve the highest possible degree of CA removal, the diafiltration process was repeated three times. The initial sample (before diafiltration) was diluted threefold with buffer, while obtained retentates in every diafiltration cycle were diluted around 50-fold (manuscript in preparation). The developed method was used to investigate the amount of CA in the initial sample (crude IgG, before diafiltration), in the filtrates after each diafiltration cycle, and in the final product (pure IgG, after diafiltration). Crude and pure IgG were prepared for HPLC analysis by precipitation with ACN in order to remove IgGs, while filtrates were submitted to HPLC analysis without any additional treatment. After the first diafiltration cycle, the majority of CA was removed (Figure 4), ending in filtrate 1 (resulting quantity was below the known CA solubility in water). In filtrate 3, as well as in the final product, the amount of CA was in the range of the LOQ value of the method. The sum of CA amounts in all filtrates and in pure IgG corresponded to the amount of CA in the initial sample, confirming the accuracy of the method and its suitability for the routine analysis of residual CA in human IgG preparations. Additionally, in the case of crude IgG diafiltration, successful removal of CA from the solution and its desorption from the membrane of the centrifugal filter unit by extensive washing were demonstrated.

## 3. Materials and Methods

### 3.1. Reagents and Chemicals

HPLC-grade acetonitrile (ACN) was purchased from Merck (Darmstadt, Germany), while HPLC-grade trifluoroacetic acid (TFA) and caprylic acid (CA) were from Sigma-Aldrich (St. Louis, MO, USA). A daily supply of ultrapure water was obtained from a PureLab Classic purification system (Elga, UK). Human Immunoglobulin for intravenous application (IVIG, 50 mg/mL) was from Institute of Immunology Inc. (Zagreb, Croatia).

### 3.2. Standard Solutions and Quality Control Sample Preparation

Concentrated stock solution (5% CA) was prepared by adding 50 μL of CA to 950 μL of ACN and filtering through a hydrophobic PTFE 0.2 μm filter (Dissolution Accessories, Oosterhout, Netherlands). The working standard solution (0.5% CA) was prepared by diluting stock solution 10-fold with mobile phase and used for further preparation of seven standard samples that covered the concentration range of 25–2500 ppm (0.0025–0.25%).

Quality control (QC) samples were prepared by spiking an IVIG solution (5 mg/mL) with an appropriate volume of 0.5% CA. In order to eliminate background noise, IVIG from QC samples was depleted prior to HPLC analysis by precipitation with one volume of ACN. Precipitates were removed by centrifugation (Eppendorf, Germany) at 3000× *g* for 45 min, and supernatants were transferred to 1.5 mL vials for HPLC analysis. Final amounts of CA in QC samples were 300, 750, and 1250 ppm. The quantification of CA was based on a CA calibration curve.

Stock solutions and samples were freshly prepared prior analysis. Each standard solution was prepared in duplicate and each QC sample was prepared in hexaplicate.

### 3.3. HPLC Instrumentation and Conditions

HPLC analysis was carried out using a Shimadzu Prominence LC-20AD with UV detector SPD-M20A, autosampler SIL20AC HT, and LabSolutions software (Kyoto, Japan). The chromatographic separation was performed on a Shim pack GIST C18 column, 2.1 × 150 mm, 3 μm (Shimadzu, Japan), fitted with a guard cartridge packed with the same stationary phase. The mobile phase consisted of a mixture of 0.05% TFA in water (A) and ACN (B) (50:50, *v*/*v*). An RP HPLC analysis was performed in isocratic mode at a flow rate of 0.2 mL/min at room temperature. A wavelength of 210 nm was used for detection. The injection volume was 5 µL, and the total run time was 10 min.

### 3.4. Validation Parameters of Developed Method

The developed HPLC method was validated according to the ICH guidelines [28]. The parameters studied for the validation included specificity, linearity, precision (repeatability and intermediate precision), accuracy, limit of detection (LOD), limit of quantitation (LOQ), and robustness.

#### 3.4.1. Specificity

The specificity of the method was assessed from chromatographic runs of standard solutions and QC samples. The retention time (RT) of CA in standard solutions and QC samples, and the separation of CA and IVIG peaks in QC samples were selected as comparison parameters.

#### 3.4.2. Linearity

In order to determine the linearity of the method, three independent calibration curves were constructed using seven standard solutions of CA prepared in duplicates, covering the range of 25–2500 ppm. Calibration curves were obtained by plotting the peak area of each standard solution as a function of amount of CA. Linearity was evaluated by linear regression analysis, using the least square linear regression method and ANOVA test in Excel (α = 0.05).

#### 3.4.3. Precision and Accuracy

Precision was determined by repeatability (intra-day precision) and intermediate precision (inter-day precision). QC samples at three different levels of CA (300, 750, and 1250 ppm) were prepared in hexaplicates (*n* = 6) and analyzed according to the proposed method (for intra-day precision and accuracy) over three consecutive days (for inter-day precision and accuracy). The precision was expressed as percentage relative standard deviation (% RSD) of the assay results, and the accuracy was expressed as the percentage recovery for each concentration of CA in QC samples according to Equation (1).
accuracy (% recovery) = (measured concentration/nominal concentration) × 100.(1)

#### 3.4.4. Limit of Detection (LOD) and Limit of Quantification (LOQ)

The limit of detection (LOD) and the limit of quantitation (LOQ) values were determined on the basis of either the signal-to-noise ratio (S/N) or a calibration curve. In both cases, analysis of standard solutions with low CA concentrations was performed. In the first approach, LOD and LOQ were determined at S/N ratios of 3:1 and 10:1, respectively.

In the second approach, LOD and LOQ were determined on the basis of calibration curves and calculated according to Equations (2) and (3).
(2)$$\mathrm{LOD}=3.3\;\frac{\sigma }{S}\;$$
(3)$$\mathrm{LOQ}=10\;\frac{\sigma }{S}$$
where *σ* is the residual standard deviation of a regression line or the standard deviation of the *y*-intercept, and *S* is the slope of the calibration curve.

#### 3.4.5. Robustness

The robustness was investigated by testing the influence of small changes in pH value, wavelength, percentage ACN content in the mobile phase, and flow rate on the chromatographic behavior of CA. The interval between low and high level for each factor was placed symmetrically around the nominal level (Table 7). In order to study the robustness, four-factor fractional factorial design (FFD) was employed (2^4−1^). The experimental design resulted in eight chromatographic runs that were performed independently (Table 8). Each run consisted of three replicate injections of QC sample. The response variables were the retention time (RT), peak area (A), theoretical plate number (N), resolution (R), capacity factor (k’), and tailing factor. TIBCO Statistica^®^ Version 14.0 (Palo Alto, CA, USA) and Microsoft Office Excel 2019 (Redmont, WA, USA) were used for the statistical evaluation of the obtained data.

## 4. Conclusions

A simple, sensitive, and rapid RP HPLC method for the determination of CA in IVIG according to the principles of good laboratory practices was successfully developed and validated under optimized conditions. The validation parameters confirmed linearity, accuracy, precision, and specificity of developed method, which provided excellent separation of CA from IgG and enabled quantification of CA in IVIG. The robustness of the developed method was evaluated using fractional factorial design. Special attention should be paid to the percentage ACN in the mobile phase and flow rate as experimental factors that have a significant impact on the chromatographic behavior of CA and, as such, should be strictly regulated during the performance of the method. The application check of the method demonstrated its appropriateness for the routine analysis of residual CA in intermediates and final products of human IgGs for passive immunotherapy in quality control.

## Figures and Tables

**Figure 1 molecules-27-01665-f001:**
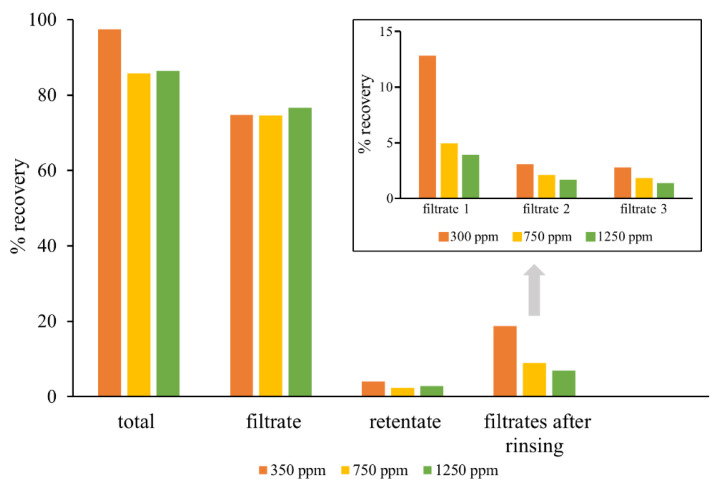
Monitoring of CA recovery in QC samples during diafiltration process by determination of its amount in filtrate, retentate, and filtrates that were obtained after three rinsing cycles with water.

**Figure 2 molecules-27-01665-f002:**
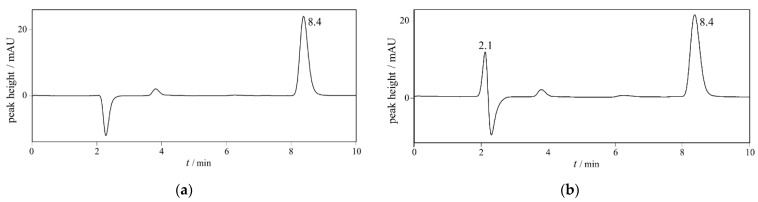
HPLC chromatograms of (**a**) caprylic acid and (**b**) QC samples (IVIG sample spiked with CA) after precipitation of IgGs by ACN. Chromatographic conditions are given in Table 1.

**Figure 3 molecules-27-01665-f003:**
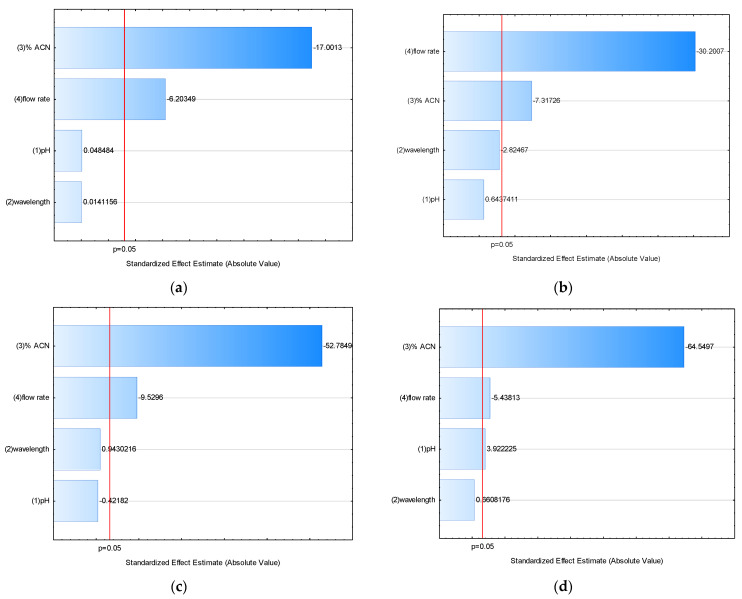
Pareto charts of standardized effects of examined operating factors (%ACN, flow rate, pH, wavelength) on responses: (**a**) retention time; (**b**) peak area; (**c**) theoretical plate number; (**d**) resolution; (**e**) capacity factor; (**f**) tailing factor.

**Figure 4 molecules-27-01665-f004:**
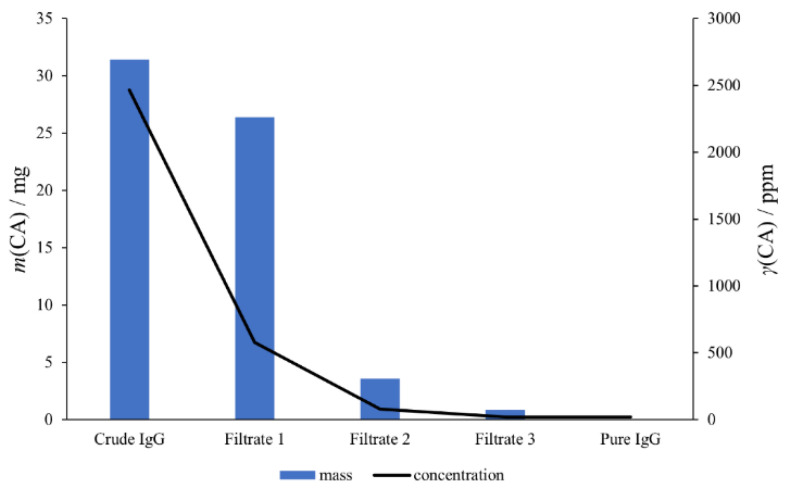
CA removal monitoring during diafiltration step of the refinement protocol for IgG purification, performed by quantification of CA amount in crude IgG, filtrates after each diafiltration step, and final pure IgG.

**Table 1 molecules-27-01665-t001:** Optimized chromatographic conditions.

Parameter	Chromatographic Condition
Mobile phase	0.05% TFA in water–ACN (50:50, *v*/*v*)
Flow rate	0.2 mL/min
Mode	Isocratic
Run time	10 min
Injection volume	5 μL
Detection wavelength	210 nm
Temperature	Ambient temperature

**Table 2 molecules-27-01665-t002:** Results of analysis of variance (ANOVA) for linearity.

	df	Sum of Squares	Mean Square	F	Significance F *
Regression	1	6.62 × 10^12^	6.62 × 10^12^	68,106.87	2.96 × 10^−66^
Residual	40	3.89 × 10^9^	97,259,462		
Total	41	6.63 × 10^12^			

* Significance F is the *p*-value of F.

**Table 3 molecules-27-01665-t003:** Precision (intra- and inter-day) and accuracy of the method for determination of CA concentration in IVIG samples.

	Nominal Amount of CA in ppm	Measured Amount of CA in ppm (Mean ± SD)	% RSD	% Recovery (Mean ± SD)
Intra-day precision ^a^				
1st day	300	303.06 ± 1.74	0.575	101.02 ± 0.58
	750	753.68 ± 5.30	0.703	100.49 ± 0.71
	1250	1249.33 ± 10.36	0.829	99.94 ± 0.83
2nd day	300	297.08 ± 1.09	0.669	99.03 ± 0.36
	750	739.37 ± 3.93	0.671	98.58 ± 0.52
	1250	1243.27 ± 7.98	0.642	99.46 ± 0.64
3rd day	300	302.31 ± 2.56	0.848	100.77 ± 0.85
	750	740.83 ± 3.38	0.457	98.78 ± 0.49
	1250	1252.12 ± 8.67	0.692	100.17 ± 0.69
Inter-day precision ^b^	300	301.04 ± 3.22	1.069	100.35 ± 1.07
	750	744.39 ± 7.64	1.027	99.25 ± 1.02
	1250	1248.24 ± 9.32	0.746	99.86 ± 0.75

^a^ *n* = 6 for each amount of CA. ^b^ *n* = 18 for each amount of CA (3 days × 6 samples).

**Table 4 molecules-27-01665-t004:** LOD and LOQ values determined by S/N ratio and SD of the response (SD of a regression line or SD of the *y*-intercept) and slope of the calibration curves.

	S/N	SD of Regression Line *	SD of *y*-Intercept *
LOD	8 ppm	10.3 ± 1.1 ppm	7.0 ± 0.7 ppm
LOQ	25 ppm	31.3 ± 3.4 ppm	21.1 ± 2.2 ppm

* Mean ± SD of calibration curves on three separate days.

**Table 5 molecules-27-01665-t005:** Values (average ± standard deviation) of obtained responses for eight experiments of fractional factorial design (FFD).

Experiment	RT	A	N	R	k’	Tailing Factor
1	12.30 ± 0.01	377.79 ± 5.43	5064 ± 54	18.38 ± 0.13	4.15 ± 0.01	1.197 ± 0.012
2	10.18 ± 0.05	312.55 ± 4.55	4632 ± 32	18.13 ± 0.02	4.21 ± 0.02	1.179 ± 0.005
3	10.17 ± 0.01	302.32 ± 4.26	4712 ± 30	17.88 ± 0.13	4.10 ± 0.01	1.176 ± 0.006
4	12.35 ± 0.03	369.27 ± 4.39	5115 ± 33	19.09 ± 0.04	4.27 ± 0.02	1.183 ± 0.005
5	6.03 ± 0.02	293.72 ± 2.93	2744 ± 24	10.64 ± 0.15	2.15 ± 0.01	1.222 ± 0.004
6	7.24 ± 0.01	358.67 ± 2.87	3029 ± 9	11.56 ± 0.07	2.17 ± 0.01	1.224 ± 0.002
7	7.23 ± 0.01	354.06 ± 2.57	3051 ± 4	11.06 ± 0.14	2.13 ± 0.01	1.234 ± 0.002
8	6.02 ± 0.01	292.91 ± 2.37	2732 ± 12	10.98 ± 0.04	2.16 ± 0.01	1.225 ± 0.004

**Table 6 molecules-27-01665-t006:** Obtained simple effect (E) of a given factor on each of the monitored responses.

Factor	E on RT	E on A	E on N	E on R	E on k’	E on Tailing Factor
pH	0.01317	1.3764	−15.92	0.44417 *	0.06942	−0.0043
Wavelength (nm)	0.00383	−6.0396	35.58	0.07483	0.00092	−0.0010
%ACN (*v*/*v*)	−4.61700 *	−15.6454 *	−1991.75 *	−7.30983 *	−2.03008 *	0.0425 *
Flow rate (mL/min)	−1.68467 *	−64.5738 *	−359.58 *	−0.61583 *	−0.02658	−0.0088

* *p* < 0.05.

**Table 7 molecules-27-01665-t007:** Experimental factors and levels.

Factors	Low Level	Nominal Level	High Level
pH	2.2	2.4	2.6
Wavelength (nm)	208	210	212
%ACN (*v*/*v*)	45	50	55
Flow rate (mL/min)	0.18	0.2	0.22

**Table 8 molecules-27-01665-t008:** The total of eight experiments proposed by the software for FFD.

Experiment Number	pH	Wavelength (nm)	%ACN (*v*/*v*)	Flow Rate (mL/min)
1	2.2	208	45	0.18
2	2.6	208	45	0.22
3	2.2	212	45	0.22
4	2.6	212	45	0.18
5	2.2	208	55	0.22
6	2.6	208	55	0.18
7	2.2	212	55	0.18
8	2.6	212	55	0.22

## Data Availability

The data presented in this study are available on request from the corresponding author.

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
