# Peer review of "Development of Improved High-Performance Liquid Chromatography Method for the Determination of Residual Caprylic Acid in Formulations of Human Immunoglobulins"

_molecules, 2022, doi:10.3390/molecules27051665_

Round 1

Reviewer 1 Report

The article entitled “Development of improved high-performance liquid chromatography method for the determination of residual caprylic acid in formulations of human immunoglobulins” is devoted to quality control of human immunoglobulin formulations. A robust and straightforward sample preparation procedure is developed to precipitate IgG and other proteins, which allowed direct HPLC-UV determination of caprylic acid. Full validation of the proposed method was performed, including an advanced robustness test by an experimental design approach. The concept of the developed method is rather simple and analytical novelty is limited; however, the use of such method will certainly improve the quality of such valuable medical preparation. There Is not much to criticize in the text and structure of the manuscript and it can be recommended to publication in the current form. The only thing, the authors could additionally discuss and explain is the difference between Standardized Effect Estimate and simple effects (E) values.

Author Response

Response to Reviewer 1 Comments

Point 1: The only thing, the authors could additionally discuss and explain is the difference between Standardized Effect Estimate and simple effects (E) values.

Response 1:  We would like to thank the Reviewer for the constructive comment. The estimated simple effect expresses the change in the respective response (in the units of response) when each operating factor is changing from low to high level. In order to identify statistically significant effects, simple effect to standard error of estimates ratio was calculated by the usage of statistical software, and such standardized effects were compared to critical values. We introduced the explanation of this difference in the manuscript.

Reviewer 2 Report

The presented paper deals with improvement of HPLC method for determination of residual caprilyc acid (CA) in human IgG formulations. The improved method involves removal of immunoglobulins by acetonitrile precipitation, prior to the determination of CA, in order to avoid interference in HPLC measurement. The improved method exhibited excellent specificity and linearity, good accuracy and precision within +-1%, and good robustness, implying only a strict control of %ACN as a crucial parameter in the method. The application of the method was demonstrated in a laboratory setup, on the example of determining residual CA during the preparation of anti-SARS-CoV-2 IgGs for immunotherapy.

The experiment is well designed and the developped method is well validated appropriate for further applications. I have two minor questions:

1. Are there any regulatives determining the allowed levels of CA in the final immunoglobuline products for human immunotherapy? It may be interesting to compare with obtained LOD and LOQ of the validated method.

2. If possible, could you briefly discuss your validation results (accuraccy, precision, LOD, LQD….) in comparison with conventional method? Are the improvements observed and how significant they are?

Author Response

Response to Reviewer 2 Comments

Point 1: Are there any regulatives determining the allowed levels of CA in the final immunoglobuline products for human immunotherapy? It may be interesting to compare with obtained LOD and LOQ of the validated method.

Response 1:  We would like to thank the Reviewer for expressing a positive opinion. To the best of our knowledge, regulatory guidelines concerning residual CA quantity in the final immunoglobulin products for human immunotherapy are still lacking. Therefore, we reformulated the sentence in the manuscript in which the regulatory requirement was mentioned. We have discussed in the manuscript that sensitivity of the method is suitable for detection of CA traces according to specifications (8 μg/mL) set for commercial products manufactured using CA precipitation as extraction step (Alonso et al. [17] and Valdenberg et al. [18]). Therefore, we find our method suitable for the control of CA amount in the final immunoglobulin products for human immunotherapy.

Point 2: If possible, could you briefly discuss your validation results (accuraccy, precision, LOD, LQD….) in comparison with conventional method? Are the improvements observed and how significant they are?

Response 2: We would like to thank the Reviewer for the suggestion. Comparison of our validation results to those obtained by the conventional method employed by Herrera et al. [11] has now been included in the manuscript. The improvements that we achieved have been highlighted according to instructions.

Reviewer 3 Report

Manuscript entitled “Development of improved high-performance liquid chromatography method for the determination of residual caprylic acid in formulations of human immunoglobulins” is original and has a significance for the scientific community in the pandemic COVID-19 . The manuscript is easy to read and materials are described in a logical and understandable way. The article is overloaded with figures (figure 4, 5) and tables.

The article needs formatting before printing.

After these minor corrections, I highly recommend to accept this manuscript for publication.

Author Response

Response to Reviewer 3 Comments

Point 1: The article is overloaded with figures (figure 4, 5) and tables. The article needs formatting before printing. 

Response 1: The authors appreciate the comment. Figure 1 and Figure 4 have now been added to Supplementary material. We would appreciate very much if the Reviewer would be willing to accept our preference for keeping other figures in the manuscript.